# SIREN: Shaping Representations for Detecting Out-of-Distribution Objects

**Xuefeng Du, Gabriel Gozum, Yifei Ming, Yixuan Li**
Department of Computer Sciences
University of Wisconsin-Madison
{xfdu,gozum,alvinming,sharonli}@cs.wisc.edu

## Abstract

Detecting out-of-distribution (OOD) objects is indispensable for safely deploying object detectors in the wild. Although distance-based OOD detection methods have demonstrated promise in image classification, they remain largely unexplored in object-level OOD detection. This paper bridges the gap by proposing a distance-based framework for detecting OOD objects, which relies on the model-agnostic representation space and provides strong generality across different neural architectures. Our proposed framework SIREN contributes two novel components: (1) a representation learning component that uses a trainable loss function to shape the representations into mixture of von Mises-Fisher (vMF) distributions on the unit hypersphere, and (2) a test-time OOD detection score leveraging the learned vMF distributions in a parametric or non-parametric way. SIREN achieves competitive performance on both the recent detection transformers and CNN-based models, improving the AUROC by a large margin compared to the previous best method. Code is publicly available at https://github.com/deeplearning-wisc/siren.

## 1 Introduction

Teaching object detectors to be aware of out-of-distribution (OOD) data is indispensable for building reliable AI systems. Today, the mainstream object detection models have been operating in the closed-world setting. That is, a model will match an object to one of the given class labels, even if it is irrelevant. Instead, the open-world setting emphasizes that objects from the unknown classes can naturally emerge, which should not be blindly predicted into a known class. In safety-critical applications, such as autonomous driving, failing to detect OOD objects on the road can directly lead to disastrous accidents [50]. The situation can be better avoided if the object detector recognizes the object as unfamiliar and appropriately cautions the human driver to take over.

In this paper, we pioneer a distance-based framework for detecting OOD objects. Currently, the distance-based method remains largely unexplored in object-level OOD detection. In particular, by operating in the representation space, distance-based methods are model-agnostic and provide strong generality across neural architectures. In contrast, existing approaches derive highly specialized OOD detection scores based on the outputs of the object detectors, which may not be seamlessly applicable across architectures. For example, the classification output of the Faster R-CNN [52] is optimized by the multi-class softmax loss, whereas the recent transformer-based object detection networks such as DEFORMABLE-DETR [75] uses multi-label focal loss [36]. Thereby, while output-based OOD scoring functions may be limited to specific architectures, distance-based methods are not.

Although distance-based OOD scoring functions have been studied in image classification, they do not trivially transfer to object detection models. For example, [33] modeled the feature embedding space as a mixture of multivariate Gaussian distributions and used the maximum Mahalanobis distance [41] to all class centroids for OOD detection. However, we observe that the modern object detection

36th Conference on Neural Information Processing Systems (NeurIPS 2022).

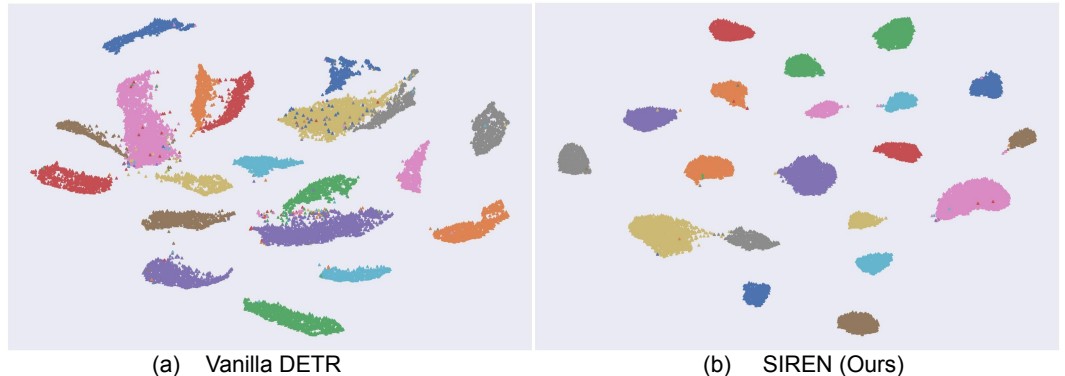

|            (a)   Vanilla DETR            |            (b)    SIREN (Ours)            |

Figure 1: (a) Feature embeddings from the penultimate layer of a vanilla DEFORMABLE-DETR [75] trained on the PASCAL-VOC dataset [14], which display irregular distributions. (b) Feature embeddings shaped by the proposed SIREN, which form compact clusters on the unit hypersphere.

models such as DEFORMABLE-DETR [75] produce highly irregular embeddings (Figure 1 (a)), which do not fit the Gaussian distributional assumption. As a result, the OOD detection score relying on such suboptimal embeddings can misbehave.

We propose a novel framework called SIREN, tackling two highly dependent problems—representation learning and OOD detection—in one synergistic framework. Concretely, SIREN contributes two novel components: **(1)** We introduce an end-to-end trainable loss that enables **Sh**ap**I**ng the **R**epres**EN**tations into a desired parametric form (Section 3.1). In particular, we model the representations by the von Mises-Fisher (vMF) distribution, a classic probability distribution in directional statistics for hyperspherical data with the unit norm. Our loss function encourages the normalized embedding to be aligned with its class prototype and shapes the overall representations into compact clusters for each class. Compared to the Gaussian distribution, using the vMF distribution avoids estimating large covariance matrices for high-dimensional data that is shown to be costly and unstable [7, 65]. **(2)** We explore test-time OOD detection by leveraging the optimized embeddings in a parametric or non-parametric way (Section 3.2). We propose a new test-time OOD score based on the learned class-conditional vMF distributions. The parameterization of the vMF distribution is directly obtainable after training, without requiring separate estimation. Different from Mahalanobis distance [33], the proposed parametric score in principle suits the learned vMF distributions on the hypersphere. Additionally, we explore a non-parametric nearest neighbor distance for OOD detection [60], which is agnostic to the type of distribution of the feature space.

Empirically, SIREN establishes superior performance on both transformer-based and CNN-based models. On PASCAL-VOC, SIREN outperforms the latest baseline OW-DETR [18] by a significant margin (↑22.53% in AUROC). Moreover, our framework is model-agnostic and does not incur changes to the existing network architecture. The proposed loss can be flexibly added as a plug-in module on top of modern architectures, as we show in Section 4.

Our key contributions are summarized as follows:

1. To the best of our knowledge, SIREN pioneers a distance-based approach for object-level OOD detection. Different from previous works, SIREN does not rely on specialized output-based OOD scores, and can generalize across different architectures in a model-agnostic fashion.

2. SIREN establishes competitive results on a challenging object-level OOD detection task. Compared to the latest method [18], SIREN improves the OOD detection performance by a considerable margin while preserving the mAP on the ID task. We show that SIREN is effective for both recent transformer-based and classic CNN-based models.

3. We shape representations via a novel vMF-based formulation for object-level OOD detection. We conduct in-depth ablations to understand how different factors impact the performance of SIREN (Section 5).

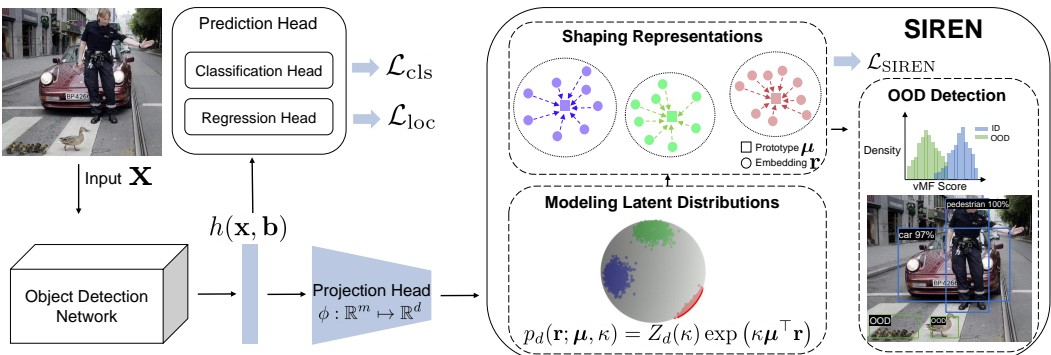

Figure 2: **Overview of the proposed learning framework** SIREN. We introduce a new loss $\mathcal{L}_{\text{SIREN}}$ which shapes the representations on the unit hypersphere into compact class-conditional vMF distributions. The embedding $\mathbf{r} \in \mathbb{R}^d$ has unit norm $\|\mathbf{r}\|^2 = 1$. In testing, we can employ either parametric or non-parametric distance functions for OOD detection. See Section 3 for details.

## 2 Preliminaries: Object-level OOD Detection

We start by introducing the OOD detection problem for object detection in the open-world setting, which has received increasing research attention lately [13, 18]. Our goal is to train object detection networks that can simultaneously: (1) localize and classify objects belonging to known categories accurately, and (2) identify unfamiliar objects outside the training categories. Compared to image-level OOD detection, object-level OOD detection is more suitable for real-world machine learning systems, yet also more challenging as it requires reasoning OOD uncertainty at the fine-grained object level. Since natural images are composed of multiple objects, knowing which regions of an image are anomalous allows for safe handling of unfamiliar objects.

**Notations.** We denote the input and label space by $\mathcal{X} = \mathbb{R}^q$ and $\mathcal{Y} = \{1, 2, ..., C\}$, respectively. Let $\mathbf{x} \in \mathcal{X}$ be the input image, $\mathbf{b} \in \mathbb{R}^4$ be the bounding box coordinates associated with objects in the image, and $y \in \mathcal{Y}$ be the semantic label of the object. An object detection model is trained on ID dataset $\mathcal{D}_{\text{tr}}^{\text{in}} = \{(\mathbf{x}_i, \mathbf{b}_i, y_i)\}_{i=1}^{M}$ drawn from an unknown joint distribution $\mathcal{P}$. We use neural networks with parameters $\theta$ to model the bounding box regression $p_\theta(\mathbf{b}|\mathbf{x})$ and the classification $p_\theta(y|\mathbf{x}, \mathbf{b})$.

**Object-level OOD detection.** The OOD detection can be formulated as a binary classification problem, distinguishing between the in- vs. out-of-distribution objects. Let $P_\mathcal{X}$ denote the marginal probability distribution on $\mathcal{X}$. Given a test input $\mathbf{x}' \sim P_\mathcal{X}$, as well as an object $\mathbf{b}'$ predicted by the object detector, the goal is to predict a binary outcome $g(\mathbf{x}', \mathbf{b}')$. We use $g = 1$ to indicate a detected object being ID, and $g = 0$ being OOD, with semantics outside the support of $\mathcal{Y}$.

## 3 Proposed Method

**Overview.** Our framework SIREN is illustrated in Figure 2, which trains an object detector in tandem with a representation-shaping branch. The object detector backbone $f : \mathcal{X} \mapsto \mathbb{R}^m$ maps an object to its feature embedding $h(\mathbf{x}, \mathbf{b}) \in \mathbb{R}^m$ (often referred to as the penultimate layer). In addition, we introduce a new MLP projection head $\phi : \mathbb{R}^m \mapsto \mathbb{R}^d$ that maps the $h(\mathbf{x}, \mathbf{b})$ to a lower-dimensional embedding $\mathbf{r} \in \mathbb{R}^d$ ($d < m$) with unit norm $\|\mathbf{r}\|^2 = 1$. The normalized embeddings are also referred to as *hyperspherical embeddings*, since they are on a unit hypersphere. In designing SIREN, we address two key challenges: **(1)** How to shape the hyperspherical representations into desirable probability distributions during training time (Section 3.1)? **(2)** How to perform test-time OOD detection by leveraging the learned distributions (Section 3.2)? Our method does not incur any change to the object detection network backbone. The proposed regularization can be flexibly used as a plug-in module on top of the modern architectures, as we will show in Section 4.

### 3.1 SIREN: Shaping Representations

**Modeling the latent distributions.** We propose to model the latent representations by the von Mises-Fisher (vMF) distribution [43], a probability distribution in directional statistics for spherical data with unit norm $\|\mathbf{r}\|^2 = 1$. The probability density function for a unit vector $\mathbf{r}$ in $\mathbb{R}^d$ is given as follows:

$$p_d(\mathbf{r}; \boldsymbol{\mu}, \kappa) = Z_d(\kappa) \exp\left(\kappa \boldsymbol{\mu}^\top \mathbf{r}\right), \tag{1}$$

where $\kappa \geq 0$, $\|\boldsymbol{\mu}\|^2 = 1$, and the normalization factor $Z_d(\kappa)$ is defined as:

$$Z_d(\kappa) = \frac{\kappa^{d/2-1}}{(2\pi)^{d/2} I_{d/2-1}(\kappa)}, \tag{2}$$

where $I_v$ is the modified Bessel function of the first kind with order $v$. $Z_d(\kappa)$ can be calculated in closed form based on $\kappa$ and the dimensionality $d$. Importantly, the vMF distribution is characterized by two parameters: the mean vector $\boldsymbol{\mu}$ and concentration parameter $\kappa$. Samples that are more aligned with the center $\boldsymbol{\mu}$ have a higher probability density, and vice versa. Here $\kappa$ indicates the tightness of the distribution around the mean direction $\boldsymbol{\mu}$. The larger value of $\kappa$, the stronger the distribution is concentrated in the mean direction. In the extreme case of $\kappa = 0$, the sample points are distributed uniformly on the hypersphere.

When considering multiple classes, we can model the embedding space as a mixture of class-conditional vMF distributions, one for each class $c \in \{1, 2, ..., C\}$:

$$p_d^c(\mathbf{r}; \boldsymbol{\mu}_c, \kappa_c) = Z_d(\kappa_c) \exp\left(\kappa_c \boldsymbol{\mu}_c^\top \mathbf{r}\right), \tag{3}$$

where $\kappa_c$ and $\boldsymbol{\mu}_c$ are class-conditional parameters. Under this probability model, an embedding vector $\mathbf{r}$ is assigned to class $c$ with the following normalized probability:

$$p(y = c | \mathbf{r}; \{\kappa_j, \boldsymbol{\mu}_j\}_{j=1}^C) = \frac{Z_d(\kappa_c) \exp\left(\kappa_c \boldsymbol{\mu}_c^\top \mathbf{r}\right)}{\sum_{j=1}^C Z_d(\kappa_j) \exp\left(\kappa_j \boldsymbol{\mu}_j^\top \mathbf{r}\right)}. \tag{4}$$

**Shaping representations.** Our key idea is to design an end-to-end trainable loss function that enables **Sh**ap**I**ng the **R**epres**EN**tations into a mixture of vMF distributions, which facilitates test-time OOD detection in the representation space (Section 3.2). We therefore name our method SIREN. The learned mapping function projects an input to a point in the embedding space, where higher probability is assigned to the correct class in comparison to incorrect classes. To achieve this, we can perform maximum likelihood estimation (MLE) on the training data:

$$\text{argmax}_\theta \prod_{i=1}^M p(y_i | \mathbf{r}_i; \{\kappa_j, \boldsymbol{\mu}_j\}_{j=1}^C), \tag{5}$$

where $i$ is the index of the object embedding and $M$ is the size of the training set. By taking the negative log-likelihood, the objective function is equivalent to minimizing the following loss:

$$\mathcal{L}_{\text{SIREN}} = -\frac{1}{M} \sum_{i=1}^M \log \frac{Z_d(\kappa_{y_i}) \exp\left(\kappa_{y_i} \boldsymbol{\mu}_{y_i}^\top \mathbf{r}_i\right)}{\sum_{j=1}^C Z_d(\kappa_j) \exp\left(\kappa_j \boldsymbol{\mu}_j^\top \mathbf{r}_i\right)}, \tag{6}$$

where $y_i$ is the ground truth label for the embedding $\mathbf{r}_i$. In effect, $\mathcal{L}_{\text{SIREN}}$ encourages the object embeddings to be aligned with its class prototype, which shapes the representations such that objects in each class form a compact cluster on the hypersphere; see Figure 1 (b).

**Prototype estimation and update.** During training, SIREN estimates the class-conditional object prototypes $\boldsymbol{\mu}_c, c \in \{1, 2, ..., C\}$. The conventional approach for estimating the prototypes is to calculate the mean vector of all training samples (or a subset of them) for each class, and update it periodically during training [74]. Despite its simplicity, this method requires alternating training and prototype estimation, which incurs a heavy computational toll and causes undesirable latency. Instead, we update the class-conditional prototypes in an exponential-moving-average (EMA) manner [34, 63]:

$$\boldsymbol{\mu}_c := \text{Normalize}(\alpha \boldsymbol{\mu}_c + (1 - \alpha)\mathbf{r}), \forall c \in \{1, 2, \ldots, C\}, \tag{7}$$

---

**Algorithm 1** SIREN: Shaping Representations for object-level OOD detection

---

**Input:** ID training data $\mathcal{D}_{\text{tr}}^{\text{in}} = \{(\mathbf{x}_i, \mathbf{b}_i, y_i)\}_{i=1}^{M}$, randomly initialized object detector and MLP projection head with parameter $\theta$, loss weight $\beta$ for $\mathcal{L}_{\text{SIREN}}$, and learnable $\{\kappa_c\}_{c=1}^{C}$.
**Output:** Object detector with parameter $\theta^*$ and OOD detector $G$.
**while** *train* **do**
    |   1. Update class-conditional prototypes $\boldsymbol{\mu}_c$ with the hyperspherical embeddings $\mathbf{r}$ by Equation (7).
    |   2. Calculate the vMF-based representation shaping loss $\mathcal{L}_{\text{SIREN}}$ by Equation (6).
    |   3. Update the learnable $\{\kappa_c\}_{c=1}^{C}$ and the network parameters $\theta$ using Equation (8).
**end**
**while** *eval* **do**
    |   1. Calculate the OOD score by Equation (9) or Equation (11).
    |   2. Perform OOD detection by Equation (10).
**end**

---

where $\alpha$ is the prototype update factor, and $\mathbf{r}$ denotes the normalized object embeddings from class $c$. The update can be done efficiently with negligible cost, enabling end-to-end training.

**Overall training objective.** The overall training objective combines the standard object detection loss, along with our new representation shaping loss $\mathcal{L}_{\text{SIREN}}$:

$$\min_{\theta, \kappa} \mathbb{E}_{(\mathbf{x}, \mathbf{b}, y) \sim \mathcal{P}} \left[ \mathcal{L}_{\text{cls}} + \mathcal{L}_{\text{loc}} \right] + \beta \cdot \mathcal{L}_{\text{SIREN}}, \tag{8}$$

where $\beta$ is the weight of our representation shaping loss. $\mathcal{L}_{\text{cls}}$ and $\mathcal{L}_{\text{loc}}$ are losses for classification and bounding box regression, respectively. We provide extensive empirical evidence in Section 4 demonstrating the efficacy of our loss function.

To the best of our knowledge, our work makes the first attempt to explore vMF-based learning and inference for object-level OOD detection. To highlight the novelty of the loss itself: we introduce a novel *learnable* $\{\kappa_j\}_{j=1}^{C}$ for each ID class in Equation (6), instead of using fixed values. Our loss allows concentration parameter $\kappa$ to adaptively and flexibly capture the class-conditional feature statistics during training. This is desirable when each ID class may have its own concentration in the hypersphere. We will later show that such learnable $\{\kappa_j\}_{j=1}^{C}$ enables better OOD detection performance (Section 5).

### 3.2 Test-time OOD Detection

During inference, we explore and contrast two types of uncertainty scores for detecting OOD objects.

**Parametric vMF score.** We propose a new test-time OOD score based on the learned class-conditional vMF distributions, parameterized by $\{\widehat{\kappa}_c, \boldsymbol{\mu}_c\}_{c=1}^{C}$. Here $\widehat{\kappa}_c$ denotes the learned concentration parameter for class $c$, which captures the concentration of representations for class $c$. For a test-time object $(\mathbf{x}', \mathbf{b}')$, we use the largest estimated class-conditional likelihood as the OOD score:

$$S(\mathbf{x}', \mathbf{b}') = \max_c Z_d(\widehat{\kappa}_c) \exp\left( \widehat{\kappa}_c \boldsymbol{\mu}_c^{\top} \mathbf{r}' \right), \tag{9}$$

where $\mathbf{r}' = \phi(h(\mathbf{x}', \mathbf{b}'))$ is the normalized embedding from the MLP projection head. Our OOD detection score thus in principle suits our learned embeddings and vMF distributions. For OOD detection, one can use the level set to distinguish between ID and OOD objects:

$$G(\mathbf{x}', \mathbf{b}') = \begin{cases} 1 & \text{if } S(\mathbf{x}', \mathbf{b}') \geq \gamma \\ 0 & \text{if } S(\mathbf{x}', \mathbf{b}') < \gamma \end{cases} \tag{10}$$

The threshold $\gamma$ can be chosen so that a high fraction of ID data (e.g., 95%) is correctly classified. For objects classified as ID, one can obtain the bounding box and class prediction using the prediction head as usual.

**Non-parametric KNN score.** To relax the distributional assumption on the learned embeddings, we additionally employ a non-parametric KNN distance for OOD detection, which performs well on compact and normalized feature space. Following Sun *et al.* [60], the KNN distance is defined as:

$$S(\mathbf{x}', \mathbf{b}', k) = -\left\| \mathbf{r}' - \mathbf{r}_{(k)} \right\|_2, \tag{11}$$

where $\mathbf{r}_{(k)}$ denotes the normalized embedding of the $k$-th nearest neighbor (in the training data), for the test embedding $\mathbf{r}'$. Our algorithm is summarized in Algorithm 1.

**Remark 1.** *Different from Mahalanobis distance [33], our parametric vMF-based OOD detection score operates under the same distributional model as the training process, and hence enjoys mathematical compatibility. Computationally, we can directly use the parameters of vMF distributions (such as $\kappa$) learned from training. In contrast, the Mahalanobis distance requires a separate test-time estimation of feature statistics—which involves an expensive and numerically unstable step of cal-*

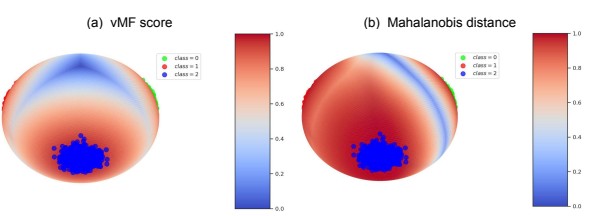

Figure 3: The uncertainty surface is calculated using our vMF score (a) and the Mahalanobis distance (b). We showcase one class for visual clarity.

*culating the covariance matrix. The distinction of the uncertainty surface calculated by both our vMF score and the Mahalanobis distance is qualitatively demonstrated in Figure 3. The data points are sampled from a mixture of three class-conditional vMF distributions (see details in Appendix F).*

## 4 Experiments

In this section, we validate the effectiveness of SIREN on object detection models, including the latest transformer-based (Section 4.1) and flagship CNN-based models (Section 4.2).

**Datasets.** Following [13], we use PASCAL-VOC[1] [14] and Berkeley DeepDrive (BDD100K)[2] [73] datasets as the ID training data. For both tasks, we evaluate on two OOD datasets that contain a subset of images from: MS-COCO [37] and OPENIMAGES (validation set) [30]. Extensive details on the datasets are in Appendix A.

**Metrics.** For evaluating the OOD detection performance, we report: (1) the false positive rate (FPR95) of OOD objects when the true positive rate of ID samples is at 95%; (2) the area under the receiver operating characteristic curve (AUROC). For evaluating the object detection performance on the ID task, we report the common metric mAP.

### 4.1 Evaluation on Transformer-based Model

**Experimental details.** We adopt a very recent DEFORMABLE-DETR (DDETR) architecture [75]. DDETR introduces multi-scale deformable attention modules in the transformer encoder and decoder layers of DETR [4], and provides better convergence and lower complexity. The multi-scale feature maps in DDETR are extracted from a ResNet-50 [21] pre-trained on ImageNet in a self-supervised fashion, *i.e.*, DINO [5]. We use the embeddings at the penultimate layer of the decoder in DDETR for projection. For the projection head, we use a two-layer MLP with a ReLU nonlinearity, with dimensionality $256 \rightarrow d \rightarrow d$. The dimension $d$ of the unit hypersphere is 16 for PASCAL-VOC and 64 for BDD100K. The default weight $\beta$ for the SIREN is 1.5 and the prototype update factor $\alpha$ is 0.95. We initialize the learnable $\kappa$ to be 10 for all classes. The $k$ in the KNN distance is set to 10. Ablations on the hyperparameters are provided in Section 5 and Appendix C. Other hyperparameters are the same as the default ones in DDETR [75].

**SIREN achieves superior performance.** In Table 1, we compare SIREN with competitive OOD detection methods in literature. For a fair comparison, all the methods only use ID data for training. SIREN outperforms competitive baselines, including Mahalanobis distance [33], KNN distance [60], CSI [61], Gram matrices [53] and Dismax [39]. These baselines operate on feature embedding space, allowing a fair comparison. Note that other common output-based methods (such as MSP [23], ODIN [35], and energy [38]) are not directly applicable for multi-label classification networks in DDETR. For these methods relying on a multi-class classification model, we will later provide comparisons on the Faster R-CNN model in Section 4.2. Implementation details and the training time for all the baseline methods are reported in Appendix D and E. We highlight a few observations:

1) The comparison between **SIREN vs. Mahalanobis** highlights precisely the benefits of our embedding shaping loss $\mathcal{L}_{\text{SIREN}}$. As shown in Figure 1, the vanilla DETR model produces ill-

---

[1]PASCAL-VOC consists of the following ID labels: Person, Car, Bicycle, Boat, Bus, Motorbike, Train, Airplane, Chair, Bottle, Dining Table, Potted Plant, TV, Sofa, Bird, Cat, Cow, Dog, Horse, Sheep.

[2]BDD100K consists of ID labels: Pedestrian, Rider, Car, Truck, Bus, Train, Motorcycle, Bicycle, Traffic light, Traffic sign.

| In-distribution dataset | Method | FPR95 ↓ | AUROC ↑ | mAP (ID)↑ |
|---|---|---|---|---|
| | | OOD: MS-COCO / OpenImages | | |
| **PASCAL-VOC** | Mahalanobis [33] | 97.39 / 97.88 | 50.28 / 49.08 | 60.6 |
| | Gram matrices [53] | 94.16 / 95.29 | 43.97 / 38.81 | 60.6 |
| | KNN [60] | 91.80 / 91.36 | 62.15 / 59.64 | 60.6 |
| | CSI [61] | 84.00 / 79.16 | 55.07 / 51.37 | 59.5 |
| | VOS [13] | 97.46 / 97.07 | 54.40 / 52.77 | 60.3 |
| | OW-DETR [18] | 93.09 / 93.82 | 55.70 / 57.80 | 58.3 |
| | Dismax [39] | 82.05 / 76.37 | 75.21 / 70.66 | 60.1 |
| | **SIREN**-vMF (ours) | 75.49±0.8 / 78.36±1.0 | 76.10±0.1 / 71.05±0.1 | 60.8±0.1 |
| | **SIREN**-KNN (ours) | **64.77**±0.2 / **65.99**±0.5 | **78.23**±0.2 / **74.93**±0.1 | 60.8±0.1 |
| **BDD100K** | Mahalanobis [33] | 70.86 / 71.43 | 76.83 / 77.98 | 31.3 |
| | Gram matrices [53] | 73.81 / 71.56 | 60.13 / 57.14 | 31.3 |
| | KNN [60] | 64.75 / 61.13 | 80.90 / 79.64 | 31.3 |
| | CSI [61] | 70.27 / 71.30 | 77.93 / 76.42 | 29.9 |
| | VOS [13] | 76.44 / 72.58 | 77.33 / 76.62 | 31.0 |
| | OW-DETR [18] | 80.78 / 77.37 | 70.29 / 73.78 | 28.1 |
| | Dismax [39] | 77.62 / 81.23 | 72.14 / 67.18 | 31.2 |
| | **SIREN**-vMF (ours) | 67.54±1.3 / 66.31±0.9 | 80.06±0.5 / 79.77±1.2 | 31.3±0.0 |
| | **SIREN**-KNN (ours) | **53.97**±0.7 / **47.28**±0.3 | **86.56**±0.1 / **89.00**±0.4 | 31.3±0.0 |

Table 1: **Main results.** Comparison with competitive out-of-distribution detection methods. All baseline methods are based on the same model backbone DDETR. ↑ indicates larger values are better and ↓ indicates smaller values are better. All values are percentages. **Bold** numbers are superior results. We report standard deviations estimated across 3 runs. SIREN-vMF/KNN denotes using vMF score and KNN distance during inference.

conditioned embeddings that do not conform to multivariate Gaussian distributions, rendering the Mahalanobis approach ineffective (with AUROC around 50%–which is random guessing). In contrast, SIREN-vMF improves the OOD detection performance (AUROC) by **25.82**% on PASCAL-VOC (MS-COCO as OOD). Different from the Mahalanobis distance, our parametric vMF scoring function naturally suits the learned hyperspherical embeddings with vMF distributions. The advantage of the vMF loss can be further verified by observing that SIREN-KNN outperforms directly applying KNN distance on the vanilla DDETR (**16.08**% AUROC improvement on VOC with COCO as OOD).

2) SIREN outperforms the latest methods VOS [13] and OW-DETR [18], which are designed for object detection models and serve as strong baselines for us. Compared with OW-DETR, SIREN-KNN substantially improves the AUROC by **22.53**% on PASCAL-VOC (COCO as OOD). OW-DETR uses the unmatched object queries with high confidence as the unknowns, and trains a binary classifier to separate ID and unknown objects. However, the unmatched object queries might be distributionally too close to the ID classes and thus displays limited improvement for OOD detection. In addition, VOS synthesizes virtual outliers from the class-conditional Gaussian distributions of the penultimate layer but fails to perform well due to the ill-conditioned embedding distribution in DDETR (non-Gaussian).

## 4.2 Evaluation on CNN-based Model

Going beyond detection transformers, we show that SIREN is also suitable and effective on CNN-based object-level OOD detection models, *e.g.*, Faster R-CNN [52]. Table 2 showcases the OOD detection performance with SIREN trained on PASCAL-VOC dataset and evaluated on both MS-COCO and OPENIMAGES datasets. In addition to baselines considered in Table 1, we include common output-based methods relying on multi-class classification, such as MSP, ODIN, and energy score.

In Table 2, we additionally report training time comparison. SIREN consistently improves OOD detection performance on both OOD datasets. Notably, SIREN performs better than

| Method | FPR95 ↓ | AUROC ↑ | Time |
|---|---|---|---|
| | COCO / OpenImages | | |
| MSP | 70.99 / 73.13 | 83.45 / 81.91 | 2.1 h |
| ODIN | 59.82 / 63.14 | 82.20 / 82.59 | 2.1 h |
| Mahalanobis | 96.46 / 96.27 | 59.25 / 57.42 | 2.1 h |
| Energy score | 56.89 / 58.69 | 83.69 / 82.98 | 2.1 h |
| Gram matrices | 62.75 / 67.42 | 79.88 / 77.62 | 2.1 h |
| KNN | 52.67 / 53.67 | 87.14 / 84.54 | 2.1 h |
| CSI | 59.91 / 57.41 | 81.83 / 82.95 | 4.9 h |
| GAN-synthesis | 60.93 / 59.97 | 83.67 / 82.67 | 3.7 h |
| VOS | 47.53 / 51.33 | 88.70 / 85.23 | 4.3 h |
| Dismax | 84.38 / 86.93 | 74.56 / 71.53 | 2.2 h |
| **SIREN**-vMF (ours) | 64.68 / 68.53 | 85.36 / 82.78 | 2.1 h |
| **SIREN**-KNN (ours) | **47.45** / **50.38** | **89.67** / **88.80** | 2.1 h |

Table 2: OOD detection results of SIREN and comparison with competitive baselines on two OOD datasets: COCO and OpenImages.

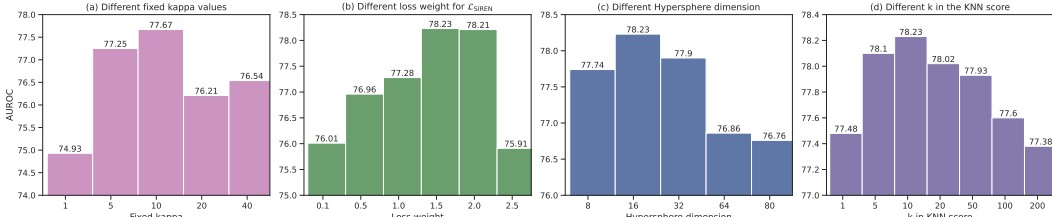

Figure 4: (a) Ablation study on using different fixed values of concentration parameter $\kappa$ in Equation (12). (b) Ablation on different weights $\beta$ for $\mathcal{L}_{\text{SIREN}}$ in Equation (8). (c) Ablation on the dimension of the hyperspherical embeddings $\mathbf{r}$. (d) Ablation study on the parameter $k$ in the KNN-based OOD detection score for SIREN. Numbers are AUROC. The ID training dataset is PASCAL-VOC and OOD dataset is MS-COCO.

the previous best OOD detection approach VOS on Faster R-CNN while preserving the same training time as vanilla Faster R-CNN.

## 5 Ablations and Discussions

In this section, we provide ablation results on how different factors impact the performance of SIREN. For consistency, we present the analyses below based on the DDETR model. Unless otherwise pointed, we use the KNN distance by default.

**Ablations on learnable vs. fixed $\kappa$.** In this ablation, we show that our approach using learnable concentration parameters $\{\kappa_c\}_{c=1}^{C}$ is better than using the fixed ones. Our method with learnable $\kappa$ is desirable, since each ID class may have its own concentration in the hypersphere. With fixed $\kappa$, the loss function can be simplified as follows:

$$\mathcal{L} = -\frac{1}{M} \sum_{i=1}^{M} \log \frac{\exp\left(\kappa \boldsymbol{\mu}_{y_i}^{\top} \mathbf{r}_i\right)}{\sum_{j=1}^{C} \exp\left(\kappa \boldsymbol{\mu}_j^{\top} \mathbf{r}_i\right)}. \tag{12}$$

Empirically, we indeed observe that employing learnable $\{\kappa_c\}_{c=1}^{C}$ achieves better OOD detection performance, with AUROC 78.23% on PASCAL-VOC model (MS-COCO as OOD). In Figure 4 (a), we show SIREN's performance when trained under different fixed $\kappa$ values. The best model under the fixed $\kappa = 10$ achieves an AUROC of 77.67%. Too small of $\kappa$ value (e.g., $\kappa = 1$) leads to almost uniform distributions and is therefore not desirable.

**Ablations on the SIREN loss weight $\beta$.** Figure 4 (b) reports the OOD detection results as we vary the weight $\beta$ for the representation shaping loss $\mathcal{L}_{\text{SIREN}}$. The model is evaluated on the MS-COCO dataset as OOD. Overall a mild weight works well. Across all the $\beta$ values considered, SIREN consistently outperforms the baseline OOD detection methods in Table 1. One can use our SIREN loss as an easy plug-in module, with minimal hyperparameter tuning.

**Ablations on the dimension $d$ of hypersphere.** SIREN projects the object feature embeddings into a lower-dimensional hypersphere in $\mathbb{R}^d$, which allows tractable vMF estimation. Figure 4 (c) shows the effect the embedding dimension $d$ on the OOD detection performance. We find that a lower dimension between 16 and 64 achieves favorable and stable performance. In the extreme case with dimension $d = 8$, the model suffers from considerable information loss and degraded performance. On the other hand, too large of $d$ causes training instability, which is not desirable either.

**Ablations on the uncertainty score.** We perform ablation on two variants of the vMF-based OOD detection score (c.f. Equation (9)): using the learned $\{\kappa_c\}_{c=1}^{C}$ vs. approximately estimate the $\kappa$ parameters directly from the converged embeddings. In literature, there are several established methods for approximating $\kappa$ [43, 55]. Denote $\bar{\mathbf{r}}_c$ the average of object embeddings for class $c$, the simplest approximate solution is given as follows:

$$\widehat{\kappa_c} = \frac{\|\bar{\mathbf{r}}_c\|(d - \|\bar{\mathbf{r}}_c\|^2)}{1 - \|\bar{\mathbf{r}}_c\|^2}, \tag{13}$$

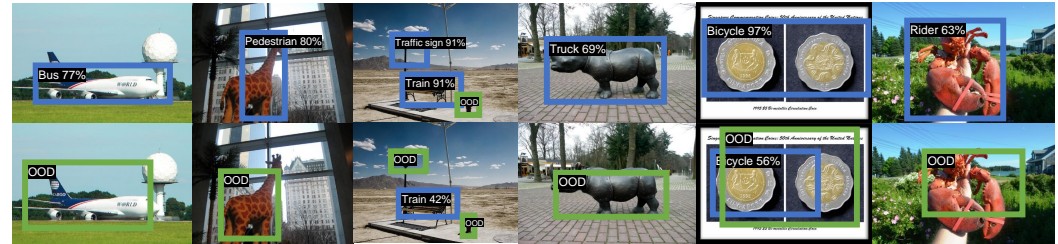

Figure 5: Visualization of detected objects on the OOD images (from MS-COCO and OPENIMAGES) by vanilla DDETR (*top*) and SIREN (*bottom*). The ID is BDD100K dataset. **Blue**: OOD objects classified as the ID classes. **Green**: OOD objects detected by DDETR and SIREN, which reduce false positives.

The proof is given in Appendix B. We show in Table 3 that using learned $\{\kappa_c\}_{c=1}^C$ avoids the imprecision in approximating $\widehat{\kappa_c}$ directly from embeddings. This affirms the importance of employing learnable concentration parameters in our SIREN loss. Moreover, the non-parametric KNN density estimation provides stronger flexibility and generality, and leads to better performance.

| | FPR95 ↓ | AUROC ↑ |
|---|---|---|
| | COCO / OpenImages as OOD | |
| vMF w/ $\kappa$ from [55] | 78.60 / 78.42 | 73.03 / 70.27 |
| vMF w/ learned $\kappa$ (ours) | 75.49 / 78.36 | 76.10 / 71.05 |
| Non-parametric KNN distance | **64.77 / 65.99** | **78.23 / 74.93** |

Table 3: Ablation on different OOD detection scores. The ID dataset is PASCAL-VOC.

**Ablations on the projection head.** We ablate on the nonlinearity in the projection head by comparing with SIREN trained with a linear layer in Table 4. The result shows using a nonlinear mapping for projection helps obtain a more expressive hypersphere, which improves OOD detection by 4.37% in terms of AUROC (MS-COCO as OOD, vMF as the OOD score).

| | FPR95 ↓ | AUROC ↑ |
|---|---|---|
| | COCO / OpenImages as OOD | |
| vMF w/o nonlinearity | 82.85 / 83.69 | 71.73 / 66.82 |
| vMF w/ nonlinearity | **75.49 / 78.36** | **76.10 / 71.05** |
| KNN w/o nonlinearity | 67.03 / 72.11 | 78.02 / 72.42 |
| KNN w/ nonlinearity | **64.77 / 65.99** | **78.23 / 74.93** |

Table 4: Ablation on the projection head. The ID dataset is PASCAL-VOC.

## 6 Qualitative analysis

In Figure 5, we visualize the predictions on several OOD images, using object detection models trained without SIREN (top) and with SIREN (bottom), respectively. The in-distribution data is BDD100K. SIREN better identifies OOD objects (in green) compared to a vanilla object detector DDETR, reducing false positives. Moreover, the confidence score of the false-positive objects of SIREN is lower than that of the vanilla model (see the train/bicycle in the 3rd/5th column).

## 7 Related work

**OOD detection for classification** can be broadly categorized into post hoc and regularization-based approaches [72]. In [2], the OpenMax score is proposed for OOD detection based on the extreme value theory (EVT). Subsequent work [23] proposed a simple baseline using maximum softmax confidence

(MSP). Several improvements have been proposed, such as deep ensemble [31], ODIN [35], distance-based score [33, 51, 53, 60], energy-based score [38, 49, 64], DICE [59] and ReAct score [58]. While post hoc methods often considered OOD scoring function alone, our framework considers both representation learning (at training) and OOD detection (at testing).

Along the line of regularization-based approaches, most prior works focused on regularizing the model's *output* [3, 16, 22, 26, 28, 42, 44, 47, 48, 62, 66, 71]. For example, the model is regularized to produce lower confidence [24, 32] or higher energy [13, 38] on the outlier data. In contrast, SIREN regularizes the model by shaping the *representations* during training. [39, 40, 61] employed self-supervised learning and learnable prototypes to learn desirable representations for novelty detection, respectively, but they used complex output-based score ensembles during inference, which are not generalizable and perform sub-optimally (see Tables 1 and 2). [57] shaped the latent representations as multivariate Gaussians but they trained a variational autoencoder and utilized the reconstruction error for open-set recognition. In contrast, SIREN directly shapes the embeddings of a *discriminative-based model*, and does not require training a generative model.

**OOD detection for object detection** is a rising topic with very few existing works. For Faster R-CNN, Du *et al.* [13] proposed to synthesize virtual outliers in the feature space for model regularization, which we compare in Section 4. [12] explored unknown-aware object detection by leveraging videos in the wild, whereas we focus on settings with still images only. For DETR, [18] adopted unmatched object queries that are with high confidence as unknowns, which did not focus on regularizing the model for desirable representations. Several works [9, 11, 19, 45, 46] used approximate Bayesian methods, such as MC-Dropout [15] for OOD detection. They require multiple inference passes to generate the uncertainty score, which are computationally expensive on larger datasets and models.

**vMF distribution in ML** has been adopted for supervised classification [29, 54], face verification [20], generative modeling [8], segmentation [25] and clustering [17], etc. Some works exploited vMF distribution for anomaly detection, they employed generative models and used the vMF distribution as the prior for zero-shot learning [6] and document analysis [76]. Both the problem settings and approaches are orthogonal to SIREN. To the best of our knowledge, our work makes the first attempt to employ vMF-based learning and inference with learnable parameters for object-level OOD detection. Metric learning has been explored for both classification [10, 67, 70, 74] and object detection [27, 56, 68, 69]. Unlike ours, they did not focus on the OOD detection problem.

## 8   Conclusion

In this paper, we propose a novel framework SIREN, which tackles object-level OOD detection with a distance-based approach. SIREN mitigates the key shortcoming of the previous output-based OOD detection approach, and explores a new vMF loss to shape representations for OOD detection. To the best of our knowledge, SIREN makes the first attempt to employ vMF-based learning and inference for OOD detection. SIREN establishes competitive performance on challenging object-level OOD detection tasks, evaluated broadly under both the recent detection transformers and CNN-based models. Our in-depth ablations provide further insights on the efficacy of SIREN. We hope our work inspires future research on OOD detection with representation shaping.

## Broader Impacts

Our project aims to improve the reliability and safety of modern machine learning models. Our study can lead to direct benefits and societal impacts, particularly for safety-critical applications such as autonomous driving. Our study does not involve any human subjects or violation of legal compliance. We do not anticipate any potentially harmful consequences to our work. Through our study and releasing our code, we hope to raise stronger research and societal awareness towards the problem of out-of-distribution detection in real-world settings.

## Acknowledgement

We thank Yiyou Sun and Ziyang (Jack) Cai for their valuable suggestions. The authors would also like to thank NeurIPS anonymous reviewers for their helpful feedback. The work is supported by the AFOSR Young Investigator Program Award.

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
