# SIREN: Shaping Representations for Detecting Out-of-distribution Objects (Appendix)

## A  Experimental Details

Following [13], we summarize the OOD detection evaluation task in Table 5. The OOD test dataset is selected from MS-COCO and OPENIMAGES dataset, which contains disjoint labels from the respective ID dataset. SIREN is trained for a total of 50 epochs on PASCAL-VOC, and trained for 30 epochs on BDD100K using ADAM optimizer. The initial learning rate is 2e-4 and decays at epoch 40 and 24 by 0.1 for PASCAL-VOC and BDD100K dataset, respectively. We set the number of the object queries as 300, the batch size as 8, and the weight $\beta$ in Equation (8) as 1.5. See *detailed ablations on the hyperparameters on a validation OOD dataset in Appendix C.*

|  | Task 1 | Task 2 |
|---|---|---|
| ID train dataset | VOC train | BDD train |
| ID val dataset | VOC val | BDD val |
| OOD dataset | COCO & OPENIMAGES val | COCO and OPENIMAGES val |
| #ID train images | 16,551 | 69,853 |
| #ID val images | 4,952 | 10,000 |
| #OOD images for COCO | 930 | 1,880 |
| #OOD images for OPENIMAGES | 1,761 | 1,761 |

Table 5: OOD detection evaluation tasks.

## B  Estimating $\widehat{\kappa}$

We provide the mathematical details for the estimation of $\kappa$ in Equation (13). Concretely, we frame the estimation problem as deriving maximum likelihood estimators (MLEs) for the vMF density function, given the training data $\{\mathbf{r}_i\}_{i=1}^M$. Specifically, the maximum likelihood learning can be written as the following optimization problem:

$$\max_{\boldsymbol{\mu},\kappa} \mathcal{L}_{\text{MLE}}(\boldsymbol{\mu},\kappa) := \sum_{i=1}^M \log p\left(\mathbf{r}_i\right) = M\kappa\boldsymbol{\mu}^\top\bar{\mathbf{r}} + M \log Z_d(\kappa) \tag{14}$$
$$\text{s.t. } \|\boldsymbol{\mu}\| = 1 \text{ and } \kappa \geq 0,$$

where $\bar{\mathbf{r}} = \frac{1}{M}\sum_{i=1}^M \mathbf{r}_i$ and $\mathcal{L}_{\text{MLE}}$ is the optimization objective of the maximum likelihood estimation for the distributional parameters $\boldsymbol{\mu}, \kappa$. In order to optimize this objective, we take the derivatives of the objective *w.r.t.* the parameters $\boldsymbol{\mu}, \kappa$ and set them to 0. Then, the optimal parameters $\boldsymbol{\mu}^*, \widehat{\kappa}$ should satisfy the following two conditions:

$$\boldsymbol{\mu}^* = \frac{\bar{\mathbf{r}}}{\|\bar{\mathbf{r}}\|}, \quad \frac{Z_d'\left(\widehat{\kappa}\right)}{Z_d\left(\widehat{\kappa}\right)} = -\|\bar{\mathbf{r}}\|, \tag{15}$$

where $Z_d'\left(\widehat{\kappa}\right) = \frac{dZ_d(\widehat{\kappa})}{d\widehat{\kappa}}$.

For the second condition in Equation (15), we let $\xi = (2\pi)^{d/2}$ and $s = d/2 - 1$ for notation simplicity. Put them into the Equation (2), we get the following:

$$Z_d(\widehat{\kappa}) = \frac{\widehat{\kappa}^s}{\xi \cdot I_s(\widehat{\kappa})}, \quad Z_d'(\widehat{\kappa}) = \frac{1}{\xi} \cdot \frac{s\widehat{\kappa}^{s-1} I_s(\widehat{\kappa}) - \widehat{\kappa}^s I_s'(\widehat{\kappa})}{I_s(\widehat{\kappa})^2}. \tag{16}$$

Divide $Z_d(\widehat{\kappa})$ by $Z_d'(\widehat{\kappa})$, we get:

$$\frac{Z_d'(\widehat{\kappa})}{Z_d(\widehat{\kappa})} = \frac{s}{\widehat{\kappa}} - \frac{I_s'(\widehat{\kappa})}{I_s(\widehat{\kappa})}. \tag{17}$$

Note that the Bessel function holds a recursive property, which is given as follows:

$$\frac{I'_s(\widehat{\kappa})}{I_s(\widehat{\kappa})} = \frac{s}{\widehat{\kappa}} + \frac{I'_{s+1}(\widehat{\kappa})}{I_s(\widehat{\kappa})}. \tag{18}$$

Substitute $\frac{I'_s(\widehat{\kappa})}{I_s(\widehat{\kappa})}$ in Equation (17) with the formula in Equation (18) and integrate it with the second condition of Equation (15), we get:

$$\frac{I_{d/2}(\widehat{\kappa})}{I_{d/2-1}(\widehat{\kappa})} = \|\bar{\mathbf{r}}\|. \tag{19}$$

Since there is no known closed-form solution to the Bessel ratio inversion problem as shown in the formula above, we adopt the approximation schemes based on the continued fraction form of the Bessel ratio function [1], namely:

$$R := \frac{I_{d/2}(\widehat{\kappa})}{I_{d/2-1}(\widehat{\kappa})} = \frac{1}{\frac{d}{\widehat{\kappa}} + \frac{1}{\frac{d+2}{\widehat{\kappa}} + \cdots}} \approx \frac{1}{\frac{d}{\widehat{\kappa}} + R}. \tag{20}$$

Replace $R$ with $\|\bar{\mathbf{r}}\|$, we get $\widehat{\kappa} \approx \frac{d \cdot \|\bar{\mathbf{r}}\|}{1 - \|\bar{\mathbf{r}}\|^2}$. Following [1], we further add a correction term $-\|\bar{\mathbf{r}}\|^3$ to the numerator and we get the approximated estimation calculated as:

$$\widehat{\kappa} = \frac{\|\bar{\mathbf{r}}\|(d - \|\bar{\mathbf{r}}\|^2)}{1 - \|\bar{\mathbf{r}}\|^2}. \tag{21}$$

## C  Hyperparameter Analysis

Below we perform sensitivity analysis for each important hyperparameter. Our sensitivity analysis uses the speckle-noised PASCAL-VOC validation dataset as OOD data, which is different from the actual OOD test datasets in use. We use SIREN pre-trained with DINO [5] as the object detection backbone, trained on in-distribution dataset PASCAL-VOC. We use the KNN score as the OOD score during inference.

**Effect of the hypersphere dimension** $d$.  SIREN projects the object feature embeddings into a lower-dimensional hypersphere in $\mathbb{R}^d$, which allows tractable vMF estimation. A reasonable choice of the hyperspherical dimension $d$ is able to preserve sufficient information for OOD detection while avoiding distributional parameter estimation in the high-dimension space. As shown in Table 6, using a dimension $d$ between 16 and 64 yields a desirable and stable performance on the OOD validation data while properly maintaining the ID performance (mAP). We set $d = 16$ for PASCAL-VOC and 64 for BDD100K in Table 1.

| $d$ | mAP↑ | FPR95 ↓ | AUROC↑ |
|---|---|---|---|
| 8 | 60.4 | 76.31 | 64.95 |
| 16 | **60.8** | **69.29** | **73.45** |
| 32 | 58.1 | 75.53 | 70.32 |
| 64 | 58.7 | 74.55 | 69.47 |
| 80 | 58.2 | 69.09 | 72.29 |

Table 6: Ablation study on the dimension of the hypersphere $d$.

**Effect of the SIREN loss weight** $\beta$.  In Table 7, we show the sensitivity of the OOD detection performance of our SIREN *w.r.t.* the weight $\beta$ of the representation shaping loss. Overall, we find that $\beta = 1.5$ achieves the best OOD detection and ID performance.

**Effect of the** $k$ **in the KNN score**.  In Table 8, we show the sensitivity of the OOD detection performance of our SIREN *w.r.t.* the $k$ of the KNN distance during inference. Overall, we find that $k = 10$ achieves the best OOD detection performance.

## D  Baselines

To evaluate the baselines, we follow the original methods in MSP [23], ODIN [35], KNN [60], and CSI [61] and apply them accordingly on the classification branch of the object detectors. The

| $\beta$ | mAP↑ | FPR95↓ | AUROC↑ |
|---|---|---|---|
| 0.1 | 60.2 | 69.72 | 73.21 |
| 0.5 | 59.2 | 71.09 | 72.36 |
| 1.0 | 59.8 | 76.20 | 70.49 |
| 1.5 | **60.8** | **69.29** | **73.45** |
| 2.0 | 58.9 | 70.41 | 71.22 |
| 2.5 | 56.0 | 77.20 | 68.37 |

Table 7: Ablation study on the loss weight $\beta$ for $\mathcal{L}_{\text{SIREN}}$.

| $k$ | mAP↑ | FPR95↓ | AUROC↑ |
|---|---|---|---|
| 1 | 60.8 | 70.42 | 72.11 |
| 5 | 60.8 | 71.66 | 72.74 |
| 10 | 60.8 | **69.29** | **73.45** |
| 20 | 60.8 | 70.03 | 72.95 |
| 50 | 60.8 | 71.85 | 72.08 |
| 100 | 60.8 | 73.73 | 71.28 |
| 200 | 60.8 | 75.96 | 70.41 |

Table 8: Ablation study on the $k$ for the KNN distance.

Mahalanobis distance [33] and gram matrices [53] are calculated based on the penultimate-layer features of the decoder in DDETR. For CSI [61], we use the rotation degree prediction (0°, 90°, 180°, 270°) as the self-supervised task. We set the temperature in the contrastive loss to 0.5. We use the penultimate-layer features of the decoder (with dimensionality 256) to perform contrastive learning. The weights of the losses that are used for classifying shifted instances and instance discrimination are both set to 1 following the original paper [61]. For OW-DETR [18], we follow the original paper and utilize the sigmoid probability of the additional unknown class for OOD detection. For VOS [13], we use the same hyperparameters as those in the original paper and synthesize virtual outliers in the penultimate layer of the object detector. Then we regard the virtual outliers as the negative samples during the object classification. For Dismax [39], we add the dismax loss with learnable prototypes in the penultimate layer of DDTER and apply the same inference score as the original paper for OOD detection.

## E    Comparison of Training Time

We provide the comparison of the training time for various baselines in Table 9 on our reported hardware (Section G in the Appendix). As the table shows, the training of our method SIREN incurs minimal computational overhead compared to the vanilla DDETR. In contrast, other baselines such as OW-DETR can be more than 2 times slower than SIREN.

| Method | Training time (h) |
|---|---|
| ID: PASCAL-VOC / BDD100K | |
| Mahalanobis [33] | 9.7 / 27 |
| Gram matrices [53] | 9.7 / 27 |
| KNN [60] | 9.7 / 27 |
| CSI [61] | 17.1 / 47.9 |
| VOS [13] | 11.4 / 32.7 |
| OW-DETR [18] | 23.7 / 59.3 |
| Dismax [39] | 10.0 / 27.7 |
| **SIREN** (ours) | 10.1 / 27.7 |

Table 9: Comparison of the training time for different baselines in Table 1 of the main paper.

## F    Details of Visualization

For Figure 3 in the main paper, we generate the toy data in the unit hypersphere by sampling from three vMF distributions in the 3D space. We adopt a concentration parameter $\kappa$ of 100 for all three

classes. The centroid vectors are set to $[0, 0, 1]$, $[\frac{\sqrt{3}}{2}, 0, -\frac{1}{2}]$ and $[-\frac{\sqrt{3}}{2}, 0, -\frac{1}{2}]$, respectively. The uncertainty surface is obtained by calculating the uncertainty score of $200^2$ points in the surface of the 3D ball.

## G   Software and Hardware

We run all experiments with Python 3.8.5 and PyTorch 1.7.0, using 8 NVIDIA GeForce RTX 2080Ti GPUs.