# OpenReview forum: "SIREN: Shaping Representations for Detecting Out-of-Distribution Objects"
_NeurIPS.cc/2022/Conference — NeurIPS 2022 Accept_

### Official Review · Reviewer_d8dM · 2022-07-03

**Rating:** 2
**Confidence:** 5
**Soundness:** 1 poor
**Presentation:** 2 fair
**Contribution:** 1 poor

**Summary:**

The paper's main assumption is that we need to construct representations during training that are compatible with the detection distributions we observe during inference.

Unlike the Mahalanobis approach, which is an inference-only solution because it does not affect the network's training, the paper proposes constructing a loss to train the network to produce consistent distribution in training and test sets.

It proposed to train the network to impose the representations to follow a mixture of von Mises-Fisher (vMF) distributions and to make both train and test distributions compatibles to improve OOD detection performance.


**Questions:**

We would like the authors to update the paper considering the comments presented on the above topics.

**Limitations:**

The paper does not properly comment on the approach's limitations. Unlike many other cited approaches, it does not allow pure end-to-end backpropagation training. It also should better explain the hyperparameter validation procedures required for the alpha, beta, and the dimension of the last layer.

**Strengths And Weaknesses:**

**Weaknesses:**

_"Shaping representations" and "leverage from the learned representations" is what loss-based approaches (IsoMax, Scaled Cosine, SNGP, DUQ, and IsoMax+) have been doing since 2019. The fact that these loss-based approaches for OOD detection were not even cited may explain why the paper understand "shaping representations" and "leverage from the learned representations" as a significant novelty. Unfortunately, it is clearly not the case._

_Unfortunately, the paper entirely ignores the last three years of relevant advances in loss-based approaches that are completely reshaping the area after the Mahalanobis inference-based approach faded out. The paper insistently compares against Mahalanabos, a four years old approach that has been outperformed very easily in the last three years._

**1. Making representation compatible with the OOD Detection procedure is at least three years old.**

The Mahalanobis paper is from 2018. At that time, inference-based metric learning approaches for OOD were trending. Since 2019, we have observed that loss-based methods are now mainstream. Therefore, training the model using a loss to make training representation and test/inference distribution compatible, which is the central claim of the paper's novelty, is not novel at all.

Unfortunately, the paper lacks to realize that training the model to produce train and inference representations compatible with performing OOD has been the mainstream approach since 2019 and was proposed by IsoMax, Scaled Cosine, DUQ, SNGP, and more recently by DisMax. Kapa appears similar to essentially the entropic scale of IsoMax.

**2. Forcing representation to the unit hypersphere is not novel.**

Forcing representation to the unit hypersphere during training to improve OOD detection was done in Scaled Cosine and IsoMax+, and more recently by DisMax.

**3. Very limited novelty.**

Considering that point #1 above showed that leveraging representations/distributions learned during training to perform OOD detection has already been proposed. Moreover, noticing that point #2 above showed that unit hypersphere representations have also already been proposed, we conclude that the novelty of the proposed approach is very limited.

**4. Neither cited nor compared with similar previous approaches.**

Considering that the paper essentially proposes a loss to train the network to improve OOD detection, we believe that the paper should have mentioned and compared with (at least) IsoMax, Scaled Cosine, DUQ, IsoMax+, and SNGP. Considering that this did not happen, we do not know if using vMF helps in any way.

**5. Results for CIFAR10 do not look impressive.**  Apparently, the results for CIFAR10 are not remarkable compared to the approaches mentioned above (a direct comparison is absolutely mandatory). We would also like to see results for CIFAR100.

**6. Results for CIFAR10 need to show classification accuracy.** We also would like to see classification accuracy results to evaluate a possible classification accuracy drop, which is very usual in loss-based approaches.

**7. We need much more experiment results using CIFAR10. Furthermore, we need CIFAR100 results as well.** CIFAR10 and CIFAR100 still are the _de facto_ standard datasets on which major OOD detection approaches have published results. The paper requires much more experimentation using CIFAR10 and CIFAR100 datasets. Classification accuracy need also to be shown and analyzed, as classification accuracy drop is a significant issue when using loss-based approaches for OOD detection.

**8. Unlike recent approaches, the proposed solutions do not allow regular pure end-to-end backpropagation-based training.**

**9. The proposed approach presents much more hyperparameters than current state-of-the-art approaches.**

**10. Results should show mean and standard deviations.**

IsoMax: https://arxiv.org/abs/1908.05569

IsoMax (journal): https://arxiv.org/abs/2006.04005

Scaled Cosine: https://arxiv.org/abs/1905.10628

DUQ: https://arxiv.org/abs/2003.02037

SNGP: https://arxiv.org/abs/2006.10108

IsoMax+: https://arxiv.org/abs/2105.14399

DisMax: https://arxiv.org/abs/2205.05874

After rebuttal: I will keep my score because the differences between the proposed model and the cited training-based approaches are not significant. No direct comparison against the methods we presented was shown. Finally, the majority of the concerns we presented were not covered in the rebuttal. Neither citing nor comparing against the previous related work for the last four years is hard to accept.

---

> ### Author Response · Authors · 2022-08-01
> **Thank you for the helpful feedback-part I**
>
> **Difference between our SIREN and loss-based approaches**
> We appreciate the reviewers for bringing our attention to the loss-based uncertainty estimation approaches! We respect the invention of these related works. However, after carefully examining these related works, we believe the main differences between them and our approach can be summarized into five folds:
>
> + First, from the motivation/novelty level, as R1 and R2 noticed, the main contribution of SIREN is that it pioneers the effort of unifying the distributional model between training and testing for OOD detection. Different from the previous works, both our representation modeling/learning and OOD detection operate under a **consistent distributional model** and enjoy strong mathematical compatibility. This is also highlighted at the end of the **Introduction** section. In contrast, the mentioned works mainly aim at designing a loss that is suitable for seamless OOD detection tasks (fast energy-efficient inferences, no classification accuracy drop, no hyperparameter tuning, and no collection of outlier data), which is not related to learning a unified and coherent distributional model for OOD detection. Additionally, even though several works, such as DisMax, employs logits with unit-norm to design losses, they did not follow the motivation of unifying the distributional model between training and testing for OOD detection. The detail of similarity calculation is also different in the unit space (L2 vs. cosine similarity).
>
> + Second, from the modeling level, SIREN estimates the class-conditional vMF distribution from data, where each class has its own learnable distributional parameters of mean and kappa, which is totally different from scaling the logits with a constant factor in IsoMax. Thus, claiming “Kapa appears similar to essentially the entropic scale of IsoMax” is not proper.
>
> + Third, from the architecture level, SIREN focuses on shaping the **intermediate representations** (such as the features of the penultimate layer) in **transformer-based** models and meanwhile demonstrate promise in CNN-based models. However, the mentioned works usually pay attention to **CNN-based models only** and manipulate the **final logit layer** in their proposed distance-based loss. We encourage the reviewer to be aware of the difference between the **representations** in SIREN and **logits** in the mentioned papers.
>
> + Moreover, from the task level, SIREN establishes state-of-the-art results on a challenging object-level OOD detection task while the mentioned works purely focus on OOD detection on classification models.
>
> + Finally, from the detail level, there are many differences between the loss-based approaches and our SIREN, such as vMF-based representation-shaping loss vs. distance-based classification loss, randomly initialized prototypes vs. prototype estimates from the vMF distributions, etc.
>
> We believe the existence of such major differences are able to support the validity and contributions of our proposed approach. At the same time, again we respect the invention of these great loss-based OOD detection approaches and are very much willing to discuss these differences after revision **with proper citations**.
>
> **Added results on the classification accuracy**
> We provide the comparison of the in-distribution accuracy of a vanilla classification network and SIREN in the following table  (evaluated on ResNet and DenseNet, respectively), which shows that SIREN will maintain the in-distribution accuracy for classification models as well.
>
> |  Method | Acc (%) |
> |:-----------:|:------------:|
> |  Vanilla cls. network  | 88.31 / 93.09  |
> |SIREN  |88.08 / 93.47  |
>
> **Added results on the CIFAR100 dataset**
> SIREN focuses on object-level OOD detection. Additionally, we provide the OOD detection performance on CIFAR100 datasets as follows  (evaluated on ResNet and DenseNet, respectively):
>
> |  Method | Acc (%) | FPR95 | AUROC|
> |:-----------:|:------------:|:------------:|:-----------:|
> |  Vanilla cls. network | 61.43 / 72.03 |  86.82 /  57.02  | 56.89 /  77.10  |
> |SIREN  |61.77 / 72.69  | **81.33 / 54.11**| **60.70 / 78.23** |
>
> **Clarification on the end-to-end backpropagation-based training**
> SIREN is **end-to-end trainable** in terms of both the object detection models and the vMF loss for shaping representations since we utilize the exponential-moving-average manner to estimate the class-conditional prototypes and the calculation formula for the distributional parameter kappa allows gradient update as well.

---

> > ### Author Response · Authors · 2022-08-01
> > **Thank you for the helpful feedback-part II**
> >
> > **Clarification on the hyperparameters**
> > Following [1], we carefully tune the hyperparameters on the speckle-noised in-distribution dataset, which are shown in **Section H** of the appendix. For the main task of object-level OOD detection, SIREN reduces the number of hyperparameters from 4 in [1] and 3 in [2] to 2 (loss weight and hyperspherical feature dimension).
> >
> > [1] Xuefeng Du, Zhaoning Wang, Mu Cai, Yixuan Li, VOS: Learning What You Don’t Know by Virtual Outlier Synthesis, ICLR2022
> >
> > [2] Akshita Gupta, Sanath Narayan, K J Joseph, Salman Khan, Fahad Shahbaz Khan, Mubarak Shah, OW-DETR: Open-world Detection Transformer, CVPR2022
> >
> > **Added mean and standard deviations**
> > For object-level OOD detection, we have provided the mean and std values in **Table 1** for SIREN. The std values for the training-based baselines are shown in the following table (evaluated on COCO and OpenImages, respectively). There are no std values for post-hoc metrics since they are deterministic.
> >
> > |  Method | mAP| FPR95 | AUROC|
> > |:-----------:|:------------:|:------------:|:-----------:|
> > |Generalized ODIN |60.4(0.7) |92.08(1.6) / 92.83(0.7) | 57.34(1.4) / 55.74(0.9)|
> > |CSI | 59.5(0.4) | 84.00(1.3) / 79.16(0.5)| 55.07(0.2) / 51.37(0.5) |
> > |VOS |60.3(0.2) |97.46(0.1) / 97.07(0.3)| 54.40(0.3) / 52.77(0.2)|
> > |OW-DETR | 58.3(0.5)|  93.09(1.2) / 93.82(0.9)| 55.70(0.8) / 57.80(0.8)|
> >
> > For the std values for the classification networks, we report on the CIFAR10 dataset (evaluated on ResNet and DenseNet, respectively).
> >
> > |  Method | FPR95 | AUROC|
> > |:-----------:|:------------:|:-----------:|
> > |Vanilla cls. network |  70.71(0.9) / 40.91(0.4) | 79.23(0.5) / 88.32(0.1)  |
> > |SIREN (ours) | 64.56(1.3) / 40.67(1.1) |  84.86(0.9) / 92.37(0.5)  |

---

> ### Comment · Reviewer_d8dM · 2022-08-10
> **After rebuttal**
>
> I will keep my score because the differences between the proposed model and the cited training-based approaches are not significant. No direct comparison against the methods we presented was shown. Finally, the majority of the concerns we presented were not covered in the rebuttal. Neither citing nor comparing against the previous related work for the last four years is hard to accept.

---

### Official Review · Reviewer_aJVX · 2022-07-10

**Rating:** 5
**Confidence:** 5
**Soundness:** 3 good
**Presentation:** 3 good
**Contribution:** 2 fair

**Summary:**

This paper proposes a novel OOD detection framework SIREN relied on the vMF distributions, which unifies representation learning and OOD detection. Detailed experiments and solid ablation studies demonstrate the effectiveness of the proposed method.

**Questions:**

1. This article select the vMF distribution to shape the image embedding, but does not clarify why the vMF distribution is the chosen one, and whether any Gaussian-like distribution is feasible?

**Ethics Review Area:**

["I don’t know"]

**Limitations:**

Limitations have been adequately addressed. No serious concerns.

**Strengths And Weaknesses:**

**Strengths**:
1. This paper is organized well and the oveall representation is sophisticated.
2. The method is plausible, and the experimental results perform well.
3. Extensive experiments and detailed ablation analysis.
4. The theoretical proof of the pivotal distribution deviation vector K has been explained in the appendix.

**Weaknesses**:
1. The originality is a bit limited, and the impact on the field is also incremental.

       Let's briefly summarize this paper, the authors introduce a pretext task of shaping the final image map, then exploit the corresponding maximum class-conditional likelihood to classify whether the current image is OOD or not. Thus, this apporach can still be coverd as an OOD method based on uncertainty and fixed thresholds.

2. Tets-time OOD dectection can hardly be highligthed as a new contribution, casue it is more like a test solution specific to this method.

3. Updating the mean of a distribution by using prototype is not fresh for me, in fact, it has appeared in the following paper:

    > Wang, Haobo, et al. "PiCO: Contrastive Label Disambiguation for Partial Label Learning." arXiv preprint arXiv:2201.08984 (2022).

---

> ### Author Response · Authors · 2022-08-01
> **Thank you for the constructive feedback**
>
> We are encouraged that the reviewer finds our work effective and plausible, performing detailed and extensive experiments and with a well-organized and sophisticated presentation. We thank the reviewer for their helpful comments and suggestions, which we address below:
>
> **Clarification on our contribution**
> Thanks for bringing our attention to the main contribution of SIREN! As noticed by R1, the main contribution of our proposed SIREN is that it pioneers the effort of shaping the representations for OOD detection under a coherent distributional model and shows effectiveness for both transformer-based and CNN-based models, which is also explained at the end of the **Introduction** section.
>
> For test-time OOD detection score, we propose to use the largest estimated class-conditional likelihood for SIREN as the OOD score, which operates under the same distributional model as the training process, and hence enjoys strong mathematical compatibility. Computationally, we can directly use the parameters of vMF distributions (such as kappa) learned from training, which makes it more stable and efficient than estimating covariance matrix in the  Mahalanobis distance. Thus, the design of the inference score serves as an important step in order to fulfill our motivation--“unifying the distributional model between training and testing for OOD detection”. The benefit of our proposed test-time inference score is illustrated in **Remark 2**.
>
> For the similar design of prototype update in other papers, absolutely fair point. We did not claim that to be our main contribution. We will discuss the design with more details in Section 3.1 and cite the relevant papers after revision, including [1, 2].
>
>
> **Clarification on the choice of vMF distribution**
> Great point raised! We choose to model the  representations by the von Mises-Fisher distribution because it is a simple and expressive probability distribution in directional statistics for hyperspherical data.
> Compared with Gaussian distribution, SIREN explores vMF distribution to estimate the distributions (**Section 3.1**), which avoids estimating large covariance matrices for high-dimensional data that is shown to be costly and unstable in literature [3,4] and **Remark 2** of our paper.  Meanwhile, choosing the vMF distribution allows us to have the features live on the unit hypersphere, which leads to several traits. For example, fixed-norm vectors are known to improve training stability in modern machine learning where dot products are ubiquitous [5], etc., which we believe is beneficial to help SIREN learn a good representation space for OOD detection.
>
> [1] Junnan Li, Caiming Xiong, Steven C.H. Hoi, MoPro: Webly Supervised Learning with Momentum Prototypes, ICLR2021.
>
> [2] Wang, Haobo, et al., PiCO: Contrastive Label Disambiguation for Partial Label Learning,  ICLR2022.
>
> [3] Kimin Lee, Kibok Lee, Honglak Lee, Jinwoo Shin, A Simple Unified Framework for Detecting Out-of-Distribution Samples and Adversarial Attacks, NeurIPS 2018
>
> [4] Xixian Chen, Michael R. Lyu, Irwin King, Toward Efficient and Accurate Covariance Matrix Estimation on Compressed Data, ICML2017
>
> [5] Tongzhou Wang, Phillip Isola, Understanding Contrastive Representation Learning through Alignment and Uniformity on the Hypersphere, ICML2020.

---

### Official Review · Reviewer_azqk · 2022-07-11

**Rating:** 5
**Confidence:** 4
**Soundness:** 2 fair
**Presentation:** 3 good
**Contribution:** 2 fair

**Summary:**

This paper shows that the representation embeddings in the transformer-based models  do not fit the Gaussian distributional assumption. As a result, previous OOD detection methods relying on such assumptions can not work.

This paper proposed the method (SIREN) to shape the representations into compact class-conditional vMF distributions, which unifies the distributional model between training and testing. And the proposed method is effective for both transformer-based and CNN-based models.


**Questions:**

Please see the "strengths and weakness".

**Limitations:**

This paper didnot describe the limitations of this work. Every paper has limitations and corner cases. I recommend the authors to supplement the limitations of this work.

**Strengths And Weaknesses:**

Strengths：

- SIREN shapes the representations for OOD detection under a coherent distributional model, which does not depend on the distribution of model representation embedding.
- SIREN unifies the distributional model between training and testing for OOD detection and shows effectiveness for both transformer-based and CNN-based models.

Weaknesses and questions：

- Why choose the vMF distributions as the coherent distributional model, rather than other distributions.

- The performance on CNN-based object detection is insufficient (table 2), and the author needs to provide more results. What's more, the VOS performs better than SIREN in CNN-based object detection, which needs some explanation.

  | Method | FPR95       | AUROC       |
  | ------ | ----------- | ----------- |
  | SIREN  | 65.45/67.33 | 84.60/81.51 |
  | VOS    | 49.02/52.11 | 88.66/85.36 |

- From figure 6, the SIREN improves the embedding quality, does this influence the performance in ID data？ And it's better to show the performance under the setting about: fixed detector and only training the projection head.

---

> ### Author Response · Authors · 2022-08-01
> **Thank you for the constructive feedback**
>
> We are glad that the reviewer found our method effective in shaping the representations for OOD detection under a coherent distributional model. We thank the reviewer for the constructive comments and suggestions, which we address below:
>
> **The performance on ID data**
> We provide the ID performance of mAP for SIREN on object detection models in Table 1, which shows that our proposed SIREN does not hurt the ID performance while significantly improves the results on OOD detection.
>
> **Clarification on the choice of vMF distribution**
> Great point raised! We choose to model the  representations by the von Mises-Fisher distribution because it is a simple and expressive probability distribution in directional statistics for hyperspherical data.
> Compared with Gaussian distribution, SIREN explores vMF distribution to estimate the distributions (**Section 3.1**), which avoids estimating large covariance matrices for high-dimensional data that is shown to be costly and unstable in literature [1,2] and **Remark 2** of our paper.  Meanwhile, choosing the vMF distribution allows us to have the features live on the unit hypersphere, which leads to several benefits. For example, fixed-norm vectors are known to improve training stability in modern machine learning where dot products are ubiquitous [3], etc., which is beneficial to help SIREN learn a good representation space for OOD detection.
>
> [1] Kimin Lee, Kibok Lee, Honglak Lee, Jinwoo Shin, A Simple Unified Framework for Detecting Out-of-Distribution Samples and Adversarial Attacks, NeurIPS 2018
>
> [2] Xixian Chen, Michael R. Lyu, Irwin King, Toward Efficient and Accurate Covariance Matrix Estimation on Compressed Data, ICML2017
>
> [3] Tongzhou Wang, Phillip Isola, Understanding Contrastive Representation Learning through Alignment and Uniformity on the Hypersphere, ICML2020.
>
> **Added results on the CNN-based object detection models**
> Another great point! The main objective of SIREN is to shape the embedding geometry for distance-based OOD detection,. Unlike VOS, SIREN does not explicitly incorporate virtual outliers for model regularization.
> In fact, we show that SIREN’s representation shaping ability can be beneficial and play a complementary role for VOS. We have tried to shape representations during training of VOS and report the OOD detection performance when we use the same OOD score, which is shown in the following table (evaluated on COCO and OpenImages, respectively). The result demonstrates the effectiveness of our training objective. We will add the comparison after revision.
>
> |  Method | FPR95 | AUROC|
> |:-----------:|:------------:|:-----------:|
> |VOS | 49.02/52.11 | 88.66/85.36|
> |VOS+SIREN |**48.01/50.28** | **89.73/86.06**|
>
> **Added results on fixed object detector**
> We report the OOD detection performance when DDETR is trained for 50 epochs without the objective of SIREN (to ensure we have a good object detector) and then fine-tune the projection head with the fixed object detector for another 50 epochs. On the Pascal VOC dataset, the comparison with SIREN is shown as follows (evaluated on COCO and OpenImages, respectively):
>
> |  Method | mAP| FPR95 | AUROC|
> |:-----------:|:------------:|:------------:|:-----------:|
> |SIREN |**60.8** | **75.49 / 78.36** |**76.10 / 71.05** |
> |SIREN w/ fixed detector | 60.5 | 83.21 / 85.94 | 69.82 / 62.07 |
>
> The table shows that training with the fixed object detector cannot outperform jointly training the projection head with the detection network. The reason may be that the joint training allows the gradients to shape representations for multiple different layers in the detection network, which is more flexible to change the representation, estimate the vMF distribution and thus is beneficial for OOD detection compared to only optimizing the MLP projection head.

---

### Author Response · Authors · 2022-08-01
**General response -- thanks to all reviewers for valuable feedback**

We are pleased to see that reviewers find that the method is **effective** and **plausible** (R1, R2), and the results on both transformer-based and CNN-based models are **detailed**, **perform well**  and **extensive** (R1, R2). We are equally glad that reviewer found the paper **organized well** and **sophisticated** (R2).

We have addressed the reviewers’ comments and concerns in **individual responses to each reviewer**. The reviews allowed us to improve our draft and the changes made in the revised draft are summarized below:

From Reviewer azqk:
+ Clarified the performance of SIREN on the ID data.
+ Added results on CNN-based object detection models and with fixed detector

From Reviewer azqk and aJVX:
+ Clarified the choice of vMF distribution.

From Reviewer aJVX and d8dM:
+ Clarified novelty and contribution

From Reviewer d8dM:
+ Added results on the classification accuracy of CIFAR100 experiments, and std values for baselines.
+ Clarification on the difference between SIREN and recent works and other confusion.

---

### Meta-Review · Area_Chair_FLd9 · 2022-08-27

**Recommendation:** Accept
**Confidence:** Less certain

**Metareview:**

This work proposes a new unified distributional model to address out of distribution detection and improves over the state of the art.

While the approach shares notable similarities with other works in the domain, the idea of creating a unified distributional representation also at the intermediate features, especially in the context of transformers is new.

While this work can be criticized for some missing comparisons with related works, the overall approach seems both sounds and novel and of general interest. Therefore, I suggest its acceptance for NeurIPS 2022.

**Award:**

No

---

### Decision · Program_Chairs · 2022-09-14

Accept